# Gene expression noise can promote the fixation of beneficial mutations in fluctuating environments

**Michael Schmutzer**[1,2], **Andreas Wagner**[1,2,3]*

**1** Department of Evolutionary Biology and Environmental Studies, University of Zürich, Zürich, Switzerland,
**2** Swiss Institute of Bioinformatics, Lausanne, Switzerland, **3** Santa Fe Institute, Santa Fe, New Mexico, USA

* andreas.wagner@ieu.uzh.ch

**Data Availability Statement:** The simulation code is publicly available from https://github.com/michaelacmschmutzer/fluctenv. Parameter values are given in the manuscript and the supporting information.

## Abstract

Nongenetic phenotypic variation can either speed up or slow down adaptive evolution. We show that it can speed up evolution in environments where available carbon and energy sources change over time. To this end, we use an experimentally validated model of *Escherichia coli* growth on two alternative carbon sources, glucose and acetate. On the superior carbon source (glucose), all cells achieve high growth rates, while on the inferior carbon source (acetate) only a small fraction of the population manages to initiate growth. Consequently, populations experience a bottleneck when the environment changes from the superior to the inferior carbon source. Growth on the inferior carbon source depends on a circuit under the control of a transcription factor that is repressed in the presence of the superior carbon source. We show that noise in the expression of this transcription factor can increase the probability that cells start growing on the inferior carbon source. In doing so, it can decrease the severity of the bottleneck and increase mean population fitness whenever this fitness is low. A modest amount of noise can also enhance the fitness effects of a beneficial allele that increases the fraction of a population initiating growth on acetate. Additionally, noise can protect this allele from extinction, accelerate its spread, and increase its likelihood of going to fixation. Central to the adaptation-enhancing principle we identify is the ability of noise to mitigate population bottlenecks, particularly in environments that fluctuate periodically. Because such bottlenecks are frequent in fluctuating environments, and because periodically fluctuating environments themselves are common, this principle may apply to a broad range of environments and organisms.

## Author summary

Genetically identical individuals that live in the same environment may differ in their behaviour and their traits. These differences arise from uncertainty inherent in all biological processes. Although this random individual variability occurs on short time scales, it can also affect evolutionary adaptation on longer time scales. For example, if a population encounters a harsh environment, random (nongenetic) differences between individuals can cause some individuals to cope better with the new environment than others. If the

**Funding:** This project has received funding from the European Research Council (https://erc.europa.eu/) under Grant Agreement No. 739874. We would also like to acknowledge support by the Swiss National Science Foundation (http://www.snf.ch/en/Pages/default.aspx) grant 31003A_172887 and by the University Priority Research Program in Evolutionary Biology (https://www.evolution.uzh.ch/en.html). The funders had no role in study design, data collection and analysis, decision to publish, or preparation of the manuscript.

**Competing interests:** The authors have declared that no competing interests exist.

ability to produce nongenetic differences is genetically determined, organisms that have this ability may have a long-term evolutionary advantage in such environments. Furthermore, if such organisms carry a beneficial gene variant, the beneficial effect of this gene variant may become amplified, and consequently spread faster in a population with more random nongenetic variation. Using a realistic model of cell growth, we show that this mechanism not only works in unfavourable environments that are stable, but also in environments that switch back and forth between a favourable and an unfavourable state. Because many natural environments undergo such periodic changes, and because random differences between individuals are ubiquitous, the mechanism we have identified may be widespread in nature.

## Introduction

Both genetic and non-genetic factors influence the rate of evolutionary adaptation. Genetic factors are well-studied, and include the intensity of natural selection [1], population size [2], standing genetic variation [3], the supply of adaptive mutations [4] and clonal interference [5]. A prominent non-genetic factor that can affect adaptation rates is variation in phenotype, variation that arises between individuals even when they share the same genome and environment [6–9].

Nongenetic phenotypic variation has the potential to either slow down or accelerate the rate of adaptation. On the one hand, nongenetic variation can lower the fitness of a population [10]. It can also cause transient random fitness differences between individuals that obscure genetic fitness differences, which weakens the effect of natural selection and increases that of genetic drift [10, 11]. Consequently, populations with a higher level of nongenetic variation may adapt more slowly, and are more prone to accumulate deleterious mutation [12].

On the other hand, nongenetic variation can increase the rate of adaptation by smoothing the fitness landscape [13]. One consequence of this smoothing is that nongenetic variation can eliminate fitness valleys in rugged landscapes, thus allowing populations to climb to higher peaks [13, 14]. Another consequence is that smoothing by nongenetic variation can make fitness gradients steeper in the vicinity of a fitness plateau [13, 15, 16]. In this case, a change in phenotype can lead to a larger increase in fitness in populations with more nongenetic variation, and the mutation underlying the novel phenotype can spread faster. Most theoretical studies, including this one, describe this second mechanism in populations that are far from their fitness optimum [13, 16, 17]. In one pertinent experiment, Bódi et al. [18] transformed yeast cells with variants of a synthetic gene circuit that induced greater or lesser variation in the expression of a gene that confers resistance to the antifungal drug fluconazole. When yeast populations harbouring these circuits were exposed to the drug, the populations with higher variation in the gene's expression not only survived higher concentrations of the drug, they also evolved drug resistance more rapidly. The reason was that high variation populations derived a greater fitness increase from beneficial mutations than low variation populations. A recent theoretical study has shown that such a mechanism can also operate close to the fitness optimum [15], but that it is more difficult to observe because nongenetic variation is generally deleterious close to the optimum.

Most work on the role of nongenetic variation in adaptation focuses on unchanging environments. With few exceptions, we know much less about this role in changing environments. The most prominent exception is the well-studied phenomenon of bet-hedging, which increases the long-term fitness of a population by minimising the population's fitness variation

in two or more environments [6, 9, 19, 20]. Through bet-hedging, organisms either display a single phenotype that performs well in multiple environments, or they display multiple phenotypes, each best suited for one environment, and switch between them [6]. Bet-hedging strategies are widespread in nature, especially among bacteria [9], but are not the focus of this study. Another exception is a theoretical study [21] predicting that in periodically fluctuating environments, genotypes underlying increased nongenetic variation can also be more sensitive to mutation, a property that may increase evolvability [22]. Another study [23] found that phenotypic plasticity, which can bring forth nongenetic variation, can protect a population against extinction when the environment suddenly changes or when it fluctuates randomly. However, in this study, the increased survival probability of a population with high plasticity did not lead to faster genetic change and evolutionary adaptation in the plastic population. In addition, a recent theoretical study has shown that even when nongenetic variation increases the mean fitness of a population, this increase does not necessarily accelerate the spreading of beneficial alleles [15]. We thus do not know whether nongenetic variation can amplify the fitness effects of a beneficial mutation in a fluctuating environment, and thus accelerate the mutation's spread through a population. To answer this question, we modelled how bacteria respond physiologically and eventually adapt evolutionarily to an environment in which the availability of a carbon and energy source changes periodically.

Our model focuses on the evolution of the lag time that bacteria need to resume growth after a change in carbon source. It pertains to an environment that switches periodically between two carbon sources. Lag times can have two (nonexclusive) causes [24]. The first, which affects all cells to a similar extent, is that cells need time to reconfigure their metabolism when the environment changes. The second is that only some cells may start to grow once a new carbon source has become available. Multiple experiments report that bacteria [24, 25] and yeast [26] split into a growing and a non-growing subpopulation after the environment switches from a superior to an inferior carbon source. If only a few cells initially grow on the new carbon source, population growth appears to stop, and only resumes again once sufficiently many cells are growing [24]. In *Escherichia coli*, this phenomenon occurs when the environment switches from glucose to acetate. Empirical observations [24] suggest that once the environment switches back to the superior carbon source glucose, all cells resume growth and there is no division of the population into growing and nongrowing subpopulations. Our model incorporates these observations by assuming that cells growing on acetate do not have a growth disadvantage in glucose. In addition, it assumes that populations with a higher fraction of cells in the growing subpopulation experience shorter lag times and have a competitive advantage. Previous theoretical studies have shown that increased gene expression noise can increase the fraction of growing cells [27], and an empirical study in yeast has demonstrated that shorter lag times can increase fitness in an environment that alternates between two carbon sources (glucose and maltose) [26]. Our model predicts that greater gene expression noise can increase the fitness of a population in fluctuating environments, augment the fitness increase derived from beneficial mutations, and accelerate the rate of evolutionary adaptation to the new environment.

## Results

### A stochastic model of carbon source switching shows a bimodal distribution of growth rates on acetate

To model a population undergoing regular fluctuations in carbon substrate, with both a growing and a nongrowing subpopulation on one substrate, we drew on a circuit in *E. coli* that controls the response to a switch in carbon source from glucose to acetate (Fig 1A). This circuit

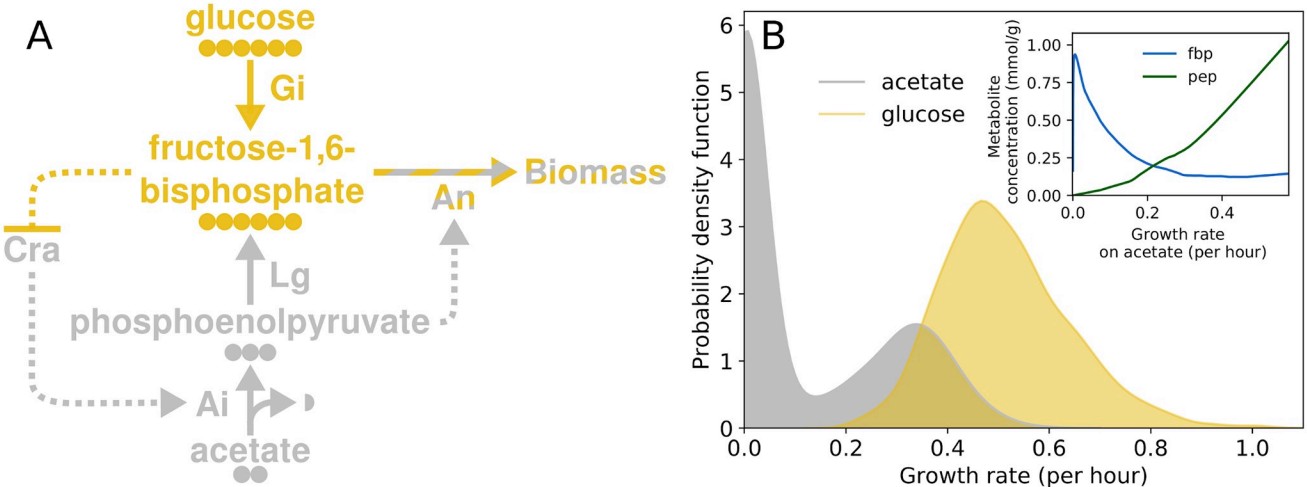

**Fig 1. Stochastic model of an *E. coli* circuit that generates a growing and a nongrowing subpopulation on acetate as well as a single growing population on glucose.** (A) Structure of the circuit. Dotted arrows show regulatory interactions and solid arrows show the flow of carbon through metabolism. Circles show the number of carbon atoms in each metabolite. The circuit has two states, a 'glycolytic' state with high fructose-1,6-bisphosphate (fbp) and low phosphoenolpyruvate (pep) concentrations (yellow) and a 'gluconeogenetic' state with low fbp and high pep concentrations (grey). Both glycolysis and gluconeogenesis eventually lead to the accumulation of biomass (striped). Carbon metabolism and growth is condensed into four reactions, each of which is catalysed by a single enzyme whose expression is noisy. Specifically, glucose and acetate incorporation and their conversion into fbp and pep are represented by enzymatic steps abbreviated as Gi and Ai respectively. Pep is converted to fbp through lower glycolysis (Lg). Growth proceeds through the conversion of fbp into biomass through anabolism (An). The transcription factor Cra, whose expression is also stochastic, is repressed by fbp and activates acetate incorporation (Ai) [24]. Pep allosterically activates An, increasing the consumption of fbp and consequently lowering its concentration. Low fbp concentrations keep Cra activity high and stabilize the uptake of acetate. In the presence of glucose, fbp concentrations are high and Cra is inhibited. (B) Predicted distribution of growth rates on glucose (yellow) and acetate (grey). When acetate is the only available carbon source, cells can be in either in the glycolytic or the gluconeogenetic state and the distribution of growth rates is bimodal. When the sole available carbon source is glucose, only the glycolytic state is active and the distribution of growth rates is unimodal. The inset shows intracellular concentrations of fbp (blue) and pep (green) in cells exposed to acetate. The concentrations were smoothed using LOWESS regression. We recorded simulated growth rates in 10 mM glucose after 720 hours of growth. To simulate the growth rates in acetate, we first equilibrated the populations for 48 hours in 10 mM glucose and recorded the growth rates after 720 hours in 20 mM acetate. We show a Gaussian kernel density estimation of the two growth rate distributions. Cra expression noise is $\eta^2 = 0.2$.

was proposed by Kotte et al. [28], has since been experimentally validated [24, 29] and plays a central role in the control of carbon metabolism [30].

In *E. coli* and many other bacteria, growth on small metabolites such as acetate requires the synthesis of larger molecules such as sugars that can serve as biosynthetic building blocks. Cells respond to a switch in carbon source from glucose to acetate by reversing the flow of metabolites through glycolysis, the main pathway for breaking down sugars [31]. The reversed pathway, known as gluconeogenesis, allows for growth on acetate [32]. The circuit that controls the switch between glycolysis and gluconeogenesis relies on the concentration of fructose-1,6-bisphosphate (fbp) as an indicator of the rate of flow through glycolysis [29]. Low fbp concentrations indicate a low rate of flow and by extension a low availability of glucose. Fbp is an inhibitor of the transcription factor Cra [33] (Fig 1A), which promotes the expression of proteins involved in acetate metabolism and gluconeogenesis [32]. Consequently, the lower the glycolytic flux, the more active Cra is, and therefore the greater the chance that a cell will start growing on acetate. Gluconeogenesis replenishes fbp, however, and in the absence of a mechanism of keeping fbp concentrations low, gluconeogenesis would come to a halt even when acetate is available. To prevent such stalling, a feedforward loop involving the metabolic intermediate phosphoenolpyruvate (pep) promotes the removal of fbp while acetate flows into a cell. Specifically, in *E. coli*, pep is an allosteric activator of fructose bisphosphatase, which removes fbp from lower glycolysis [33] and thus plays an important role in stabilizing gluconeogenesis in the presence of acetate [24]. If insufficient pep is produced, fbp concentrations

can remain high enough to inhibit Cra, and the cell becomes locked in a state with no growth even if plenty of acetate is available. In contrast to growth on acetate, all cells are capable of growing on glucose, regardless of whether they previously grew on acetate or not.

As a result of the circuit's regulatory dynamics, cells can be in two states on acetate that are distinguished by the concentrations of fbp and pep (Fig 1B). One state involves a high fbp and a low pep concentration. Because fbp inhibits Cra, acetate metabolism remains low, and the pep concentration is too low to promote the removal of fbp. Cells in this state do not grow. The other state involves a high pep concentration and a low fbp concentration, and therefore a high Cra activity that promotes the intake of acetate and its conversion into pep. Cells in this state do grow. Cells switch between states depending upon how active Cra is. Cra activity depends on the number of Cra molecules within a cell, the fbp concentration, and the strength of binding between Cra and fbp. In the presence of glucose, fbp concentrations are high and acetate metabolism is repressed. These two states can arise even in a purely deterministic regulatory circuit that does not involve stochastic fluctuations of molecular concentrations (see S1 Appendix section 6), and it is to some extent robust to changes in the values of biochemical parameters (S1 Appendix section 7). We modelled this deterministic circuit with a system of three ordinary differential equations (ODEs) that determine the temporal change in pep and fbp concentration in each cell (Eqs 2 and 3 respectively), and the resultant change in cell mass (Eq 1).

Because stochastic fluctuations are in practice important for the switching between states, we embedded the deterministic model into a more comprehensive model that includes stochastic protein production (Fig 1A). In this model, we condensed central metabolism into four reactions that link the uptake of acetate or glucose to the production of biomass (Fig 1A). Each reaction is catalysed by a single protein enzyme whose expression is stochastic. These proteins are expressed constitutively except for the protein involved in acetate uptake, whose expression is activated by Cra. Because only few proteins are actively degraded in *E. coli* [34], we set the rate of active protein degradation to zero, an assumption whose consequences we assess in the supporting information (S1 Appendix section 9). Because the number of proteins in a cell fluctuates randomly, the rate of the reactions catalysed by these proteins will fluctuate as well, and this random variation in reaction rates affects the cell growth rate. Also, fluctuations in the number of Cra molecules affect whether cells occupy the growing or nongrowing state on acetate (see S1 Appendix section 8). Cells start with a fixed amount of biomass, and divide once their mass has doubled.

Our model permits us to tune gene expression noise while keeping the mean protein number per cell constant. In nature, mutations affect both the mean and the variance of gene expression levels. Uncoupling these two is important to study the effect of gene expression noise on fitness independently from that of mean expression. Also, theory suggests that mean and variance can evolve independently during adaptive evolution [35]. In addition, they can be uncoupled experimentally [36]. We quantify expression noise ($\eta^2$) as the squared coefficient of variation $\sigma^2/\bar{x}^2$, where $\bar{x}$ and $\sigma$ are the mean and standard deviation of a protein's concentration (see [37], as well as Materials and Methods for details on the model underlying gene expression noise, and S1 Appendix section 3, for the parameter values underlying each level of noise). Specifically, we investigated four levels of Cra expression noise that span four orders of magnitude from $10^{-2}$ to $10^1$, and which correspond to a standard deviation of 14 to 316 Cra molecules per cell for a mean of 100 Cra molecules per cell. This concentration is low for a global regulator [38], but consistent with the slow growth conditions we simulate here [37, 39]. These levels of noise are within the range observed in *E. coli* except for the lowest ($10^{-2}$). However, even this low value can be observed *in vivo* for synthetic combinations of promoters and ribosome binding sites [40, 41].

In an environment containing glucose as the only carbon source, the model results in a unimodal distribution of growth rates on glucose. In an environment containing only acetate it reveals a bimodal distribution of growth rates (Fig 1B). To assess how often cells switch between the growing and the nongrowing state in acetate, we simulated 1000 cells growing on 20 mM acetate for 2100 hours, and followed only one of the offspring cells of each cell division (that is, we followed a single cell lineage). We observed that cell lineages remain in the non-growing state for about 110 hours (± 210 hours standard deviation), and that they remain in the growing state for about 450 hours (± 500 hours). Thus, once cells are locked in a given growth state on acetate, switching to the other state is rare.

## Populations with greater Cra expression noise have shorter lag times after a switch to acetate

Because cells with more Cra molecules are more likely to grow in acetate (S1 Appendix section 8), we investigated how Cra expression noise affects lag times after a switch from glucose to acetate. Increased Cra expression noise increases the likelihood that cells arise which host very large numbers of Cra molecules. These cells may have an elevated probability of initiating growth on acetate, and if they do, their descendants may drive population growth. We reasoned that by increasing the proportion of such rare cells, Cra expression noise may decrease a population's lag time and thus increase the mean population fitness.

We simulated the population dynamics of four kinds of populations, which differed in their level of Cra expression noise. These populations grew in a simulated environment through which nutrient medium flowed at a constant rate. The constant flow of medium flushed out cells at random, and the concentration of a carbon source (glucose or acetate) in the growth environment depended on its influx, efflux and uptake by cells. Each population started growing from an initial number of 2000 cells and experienced a single change from glucose to acetate after 48 hours of growth. We observed that populations with high Cra expression noise resume growth earlier and experience less severe bottlenecks in population size after the shift to acetate (Fig 2A). Specifically, the size of the noisiest populations (Fig 2B) decreases to an average of 1500 cells (± 80 cells), whereas the size of the least noisy populations decreases to an average of 580 cells (± 90 cells). The smallest population sizes differ significantly between populations with different levels of Cra expression noise (pairwise Wilcoxon rank sum test with Holm's correction, $p < 0.001$, except for populations with the two lowest levels of noise, which are not significantly different $p = 0.064$). The least noisy populations also take—on average— twice as long as the noisiest populations to rebound and reach carrying capacity (we use the term 'carrying capacity' in the sense of equilibrium population size), from 17 hours (± 1 hour) after the switch to acetate for the populations with the highest noise to 33 hours (± 2 hours) for the population with the lowest noise (Fig 2C). Overall, populations with lower Cra expression noise take significantly longer to rebound and reach carrying capacity in acetate (One-Way ANOVA, F = 734.9, degrees of freedom 3 and 196, $p < 0.001$). In sum, populations with noisier Cra expression experience shorter lag times on average.

## Cra expression noise increases fitness

To quantify how much fitness increases with shorter lag times, we simulated a competitive fitness assay. For each of our four levels of Cra expression noise, we competed 50 replicate populations against a reference population with an intermediate level of Cra expression noise ($\eta^2 = 0.2$). We started this in silico competition with both competitors occurring at an initial frequency of 50% and a total of 2000 cells. As a measure of fitness, we quantified the number of descendants after a total of four days of growth and a single shift from glucose to acetate half-

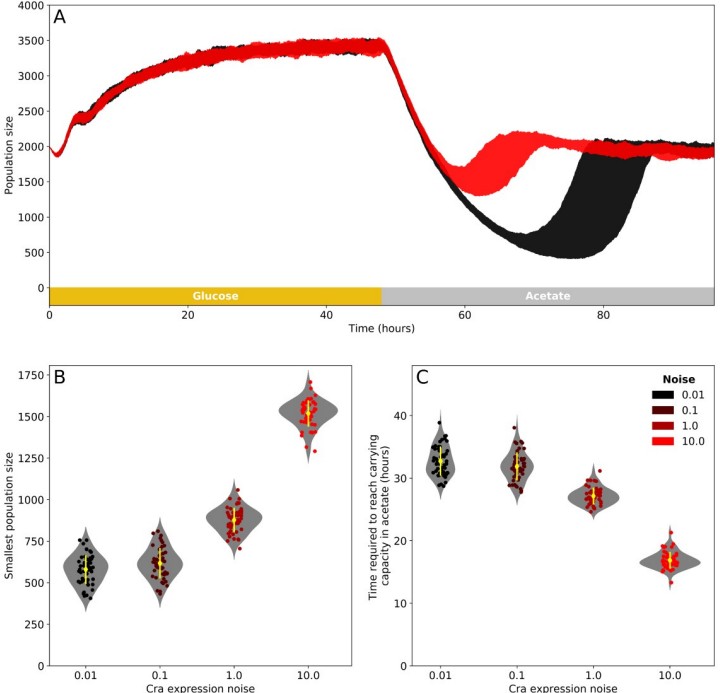

**Fig 2. Cra expression noise improves population recovery after a switch from glucose to acetate.** (A) Population size during two days of growth in glucose (yellow box) followed by two days in acetate (grey box). Growth occurs in an environment with constant medium flow that flushes cells out at a continuous rate. The areas shaded in red (highest Cra expression noise) and black (lowest Cra expression noise) show the range in population sizes observed at every point in time for 50 replicate simulations. After the switch in carbon source, populations collapse and recover to carrying capacity in acetate. For clarity, we only show results from the noisiest and least noisy populations in this panel. (B) Smallest population size observed in acetate. (C) Time required to reach carrying capacity after the switch to acetate. In (B) and (C), the yellow diamonds show the mean and the yellow lines one standard deviation. Circles show observations for each of the 50 replicates and are coloured by the level of Cra expression noise. Grey violin plots are Gaussian kernel estimates of the distribution of the data.

way during this time (see Materials and methods). Increasing Cra expression noise improves fitness (Fig 3A). Compared to the reference population, the least noisy population is on average slightly less fit than the reference population (relative fitness of -0.018 ± 0.085, zero indicates no fitness difference), while the noisiest population is substantially fitter (0.177 ± 0.056). These fitness differences between populations with different levels of noise are significant (one-way ANOVA, F = 59.8, degrees of freedom 3 and 196, $p < 0.001$). Only the fitness values between populations with the lowest levels of Cra expresssion noise 0.01 and 0.1 are not significantly different from each other (Tukey's HSD, $p = 0.312$).

We also quantified how Cra expression noise changes the fitness landscape. To create this fitness landscape, we repeated the in silico competitions with populations that not only varied in the level of noise in Cra expression, but also in the duration of the lag time, which we modified by changing the strength of binding between Cra and its inhibitor fbp. Looser binding increases Cra activity and consequently the probability that cells will grow on acetate, thus shortening lag times and increasing fitness. We modulated the Cra-fbp binding strength by changing the Cra-fbp dissociation constant (the higher the dissociation constant, the weaker binding is). We competed each of these populations against a single reference population that has intermediate values for both Cra expression noise and Cra-fbp binding strength (Fig 3B). While noisier populations have a clear fitness advantage when Cra-fbp binding was strong,

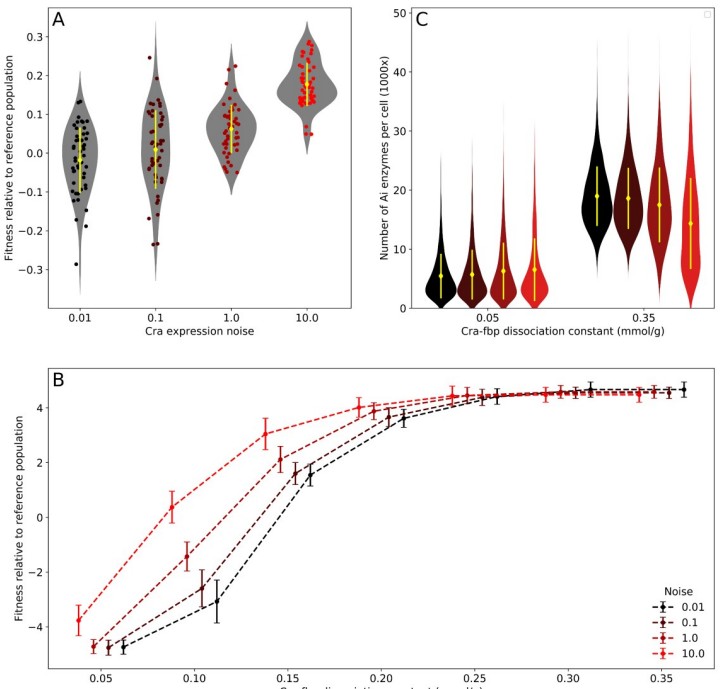

**Fig 3. Cra expression noise increases fitness in a fluctuating environment when a population has low mean fitness.** (A) Mean population fitness increases with Cra expression noise. We estimated fitness values by competing a population against a reference population with an intermediate level of Cra expression noise ($\eta^2 = 0.2$) and the same Cra-fbp dissociation constant as the reference (0.1 mmol g$^{-1}$), with 50 replicate simulations for each competition. In each replicate, both populations started with initially 1000 cells and grew together for four days in an environment with a constant flow of medium where the carbon source changed from glucose to acetate after two days. We estimated fitness from the change in the relative frequency of the non-reference population. Yellow diamonds and lines show the sample mean and one standard deviation. Circles represent the fitness values observed in each replicate, and grey violin plots are Gaussian kernel density estimates of the distribution of fitness values. (B) The fitness benefit derived from looser Cra-fbp binding is greater in populations with noisier Cra expression when the mean population fitness is low. We note that lower Cra-fbp dissociation constants correspond to greater binding strength. In this panel, fitness was determined through competition with a reference population that had both intermediate Cra expression noise ($\eta^2 = 0.2$) and an intermediate Cra-fbp dissociation constant (0.125 mmol g$^{-1}$), with 50 replicates for each competition. Circles denote mean fitness and are slightly offset on the horizontal axis for clarity. Error bars show one standard deviation. (C) Relaxing Cra-fbp binding increases the expression of the acetate incorporation enzyme Ai. We simulated cell growth on acetate in an environment with continuous medium through-flow for 21 days to equilibrate the population and then recorded the number of proteins per cell. Yellow diamonds and lines show the sample mean and one standard deviation. The violin plots are a Gaussian kernel density estimate of the distribution of proteins. The number of cells sampled for each combination of noise and Cra-fbp dissociation constant are 1637, 1619, 1835, 1719, 2005, 1991, 1981, and 2020.

this advantage disappears as Cra-fbp binding loosens. Cra expression noise does not become detrimental at high fitness, however. Instead, weak Cra-fbp binding causes Cra to have high activity in all cells exposed to acetate, which leads to a reliably high expression of the acetate uptake enzyme throughout the population (Fig 3C).

## Gene expression noise accelerates the spread of a beneficial allele by modulating the ratio of bottleneck sizes

In the previous two sections we have shown that Cra expression noise decreases the lag time, mitigates the severity of the population bottleneck in acetate (Fig 2A and 2B), and therefore increases fitness. In this section we will investigate whether Cra expression noise can accelerate

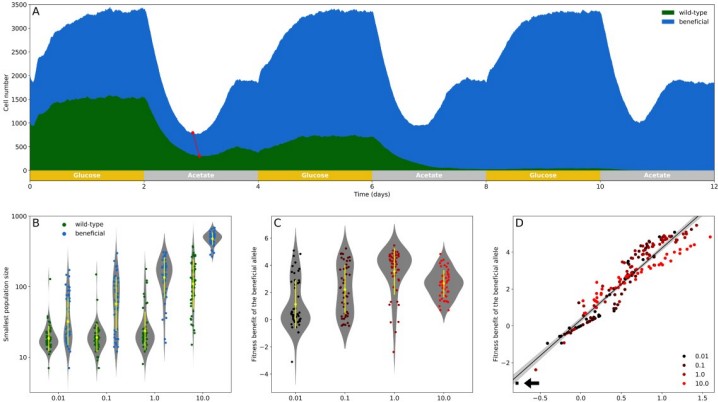

**Fig 4. The ratio of the smallest number of cells carrying different Cra alleles in acetate is a strong predictor of the fitness of cells carrying the beneficial allele.** (A) Stackplot of the number of cells carrying either the wild-type or beneficial allele (dissociation constant 0.05 and 0.07 mmol $g^{-1}$, in green and blue respectively) growing in an environment that switches between glucose and acetate every two days. We show an example simulation in which both subpopulations have a Cra expression noise of $\eta^2 = 10$. All simulations start with 2000 cells, where 1000 cells carry either allele. The two connected red dots show the smallest number of cells carrying either allele during the first exposure to acetate. (B) Smallest number of cells for both competing alleles during the first exposure to acetate. (C) Fitness of the beneficial allele relative to the wild-type allele as estimated from the change in relative frequency in the four days since the beginning of the simulation. (D) The fitness benefit of the subpopulation carrying a beneficial Cra allele correlates strongly with the logarithm of the ratio between its smallest population size in acetate and that of the subpopulation carrying the wild-type allele (Pearson's $r = 0.95$, $p < 0.001$, the line of best fit is shown in black with the 95% confidence interval in grey). Circles show the values observed in 50 replicates. The black square (arrow) shows the one replicate for a Cra expression noise of $\eta^2 = 0.01$ where the beneficial allele was lost from the population. For (B) and (C), yellow diamonds and lines show the sample mean and one standard deviation, and every circle is an observation from a replicate observation. All plots show the result of 50 replicates for each level of Cra expression noise.

the spread of a beneficial allele that also decreases lag times. This beneficial variant loosens Cra-fbp binding by increasing the Cra-fbp dissociation constant, thus increasing Cra activity.

Previous theoretical work has established that selective bottlenecks, i.e. bottlenecks in which two competing genotypes differ in their ability to survive a bottleneck, can accelerate adaptation [42]. The greater the relative advantage of a genotype in bottleneck survival, the faster the genotypes reaches fixation, especially when the bottleneck is severe. We therefore investigated whether Cra expression noise influences how effective the beneficial allele is at mitigating the population bottleneck in acetate. We then examined the effect of these population dynamics on the fitness of the beneficial allele before considering the rate at which the beneficial allele spreads through a population.

We simulated populations in which all cells have the same Cra expression noise, but initially 50% of the cells in a population of 2000 cells carry the beneficial allele, and the remainder carry a wild-type allele. We chose both alleles from the region of the fitness landscape with the lowest fitness (Fig 3B). In these simulations, the environment changed in carbon source between glucose and acetate every two days, and the simulations continued until one of the two alleles had gone to fixation (Fig 4A). We quantified the fitness of the beneficial allele relative to the wild-type allele from the change in allele frequency after four days and for 50 replicate simulations. We also recorded how much time it took for the beneficial allele to become fixed. Indeed, the shortened lag times (Fig 4B) of the beneficial allele give cells a competitive advantage (Fig 4C).

When we studied how the fate of the beneficial allele depends on gene expression noise, we first found that it reduces the severity of population bottlenecks more in populations with higher Cra expression noise. We quantified the smallest population size during the first

exposure to acetate for both subpopulations (Fig 4A, red circles). As Cra expression noise increases, both competing subpopulations experience less severe bottlenecks after the switch to acetate (Fig 4B). The smallest population size increases more for the subpopulations with the beneficial allele. For example, in the populations with the lowest Cra expression noise, the smallest population sizes differ on average by 12 cells (± 33 cells), while in the populations with the highest Cra expression noise, the average difference in population size is 177 cells (± 202 cells). Gene expression noise thus magnifies the advantage derived from the beneficial allele.

Second, we did observe an increasing fitness advantage of the beneficial allele with greater Cra expression noise, but only up to a point. The average fitness benefit (Fig 4C) increases with noise from 1.13 (± 1.80) up to 3.53 (± 1.7) for a Cra expression noise of $\eta^2 = 1$ (ANOVA, F = 18.9, degrees of freedom 3 and 196, $p < 0.001$). For populations with the largest Cra expression noise ($\eta^2 = 10$), we found a lower fitness gain (2.65 ± 0.99) from the beneficial allele than for populations with the second largest level of noise (Tukey's HSD test, $p = 0.035$ for the pairwise comparison of populations with Cra expression noise 10 and 1). This reduction in the relative fitness benefit occurs because the highest level of Cra expression noise also substantially increases the smallest cell count of the subpopulation carrying the wild-type allele (Fig 4B). In contrast, the lower levels of Cra expression noise have only a small impact on the smallest cell count of subpopulations carrying the wild-type allele, but affect subpopulations carrying the beneficial allele much more (Fig 4B). Consequently, the factor determining the outcome of competition is not the difference but the ratio between the smallest populations sizes of the two subpopulations. This ratio correlates strongly with the fitness benefit derived from the beneficial allele (Pearson's $r = 0.95$ with $p < 0.001$, Fig 4D). In sum, the relative fitness benefit of the beneficial allele increases with Cra expression noise, but only as long as the benefit of noise for the wild-type allele is slight in comparison.

Third, we examined whether Cra expression noise, given that it can increase the fitness benefit of a beneficial allele, can also accelerate the spread of the beneficial allele through a population and shorten the time to fixation of the beneficial allele. For this analysis we quantified the change in the frequency of the beneficial allele during the simulation, and recorded the fixation time. The greatest change in allele frequency occurs during periods of acetate exposure (Fig 5A and 5B), and changes in allele frequency become less predictable with lower Cra noise. In one population with the least noise, the beneficial allele is even lost from the population (Fig 5B). The beneficial allele tends to go to fixation the fastest in populations with the second highest level of Cra expression noise, with the fixation time falling from an average of 12 days (± 5 days) at the lowest noise to 9 days (± 2 days) for the second highest level of noise $\eta^2 = 1$ (pairwise Wilcoxon rank sum test with Holm's correction, $p = 0.002$ for the pairwise comparison of fixation times in populations with Cra expression noise $\eta^2 = 1$ and 0.1, or else $p < 0.001$ between $\eta^2 = 1$ and all other populations). The beneficial allele spreads considerably slower in populations with the highest Cra expression noise (11 days ± 2 days), and the fixation times are not significantly different from those of populations with the lowest two Cra expression noises ($p = 0.357$ for the pairwise comparison of $\eta^2 = 10$ and 0.1, and $p = 0.082$ for $\eta^2 = 10$ and 0.01). The fixation time is well explained by the relative fitness of these populations as quantified during the first four days of growth. The greater the difference in fitness between the two alleles, the less time it takes for the fitter allele to go to fixation (Spearman's $\rho = -0.77$ with $p < 0.001$, Fig 5D).

Taken together, these observations show that moderate levels of Cra expression noise can increase the fitness advantage of a beneficial allele during a selective bottleneck. Consequently, the beneficial allele reaches fixation faster in populations with intermediate Cra expression noise.

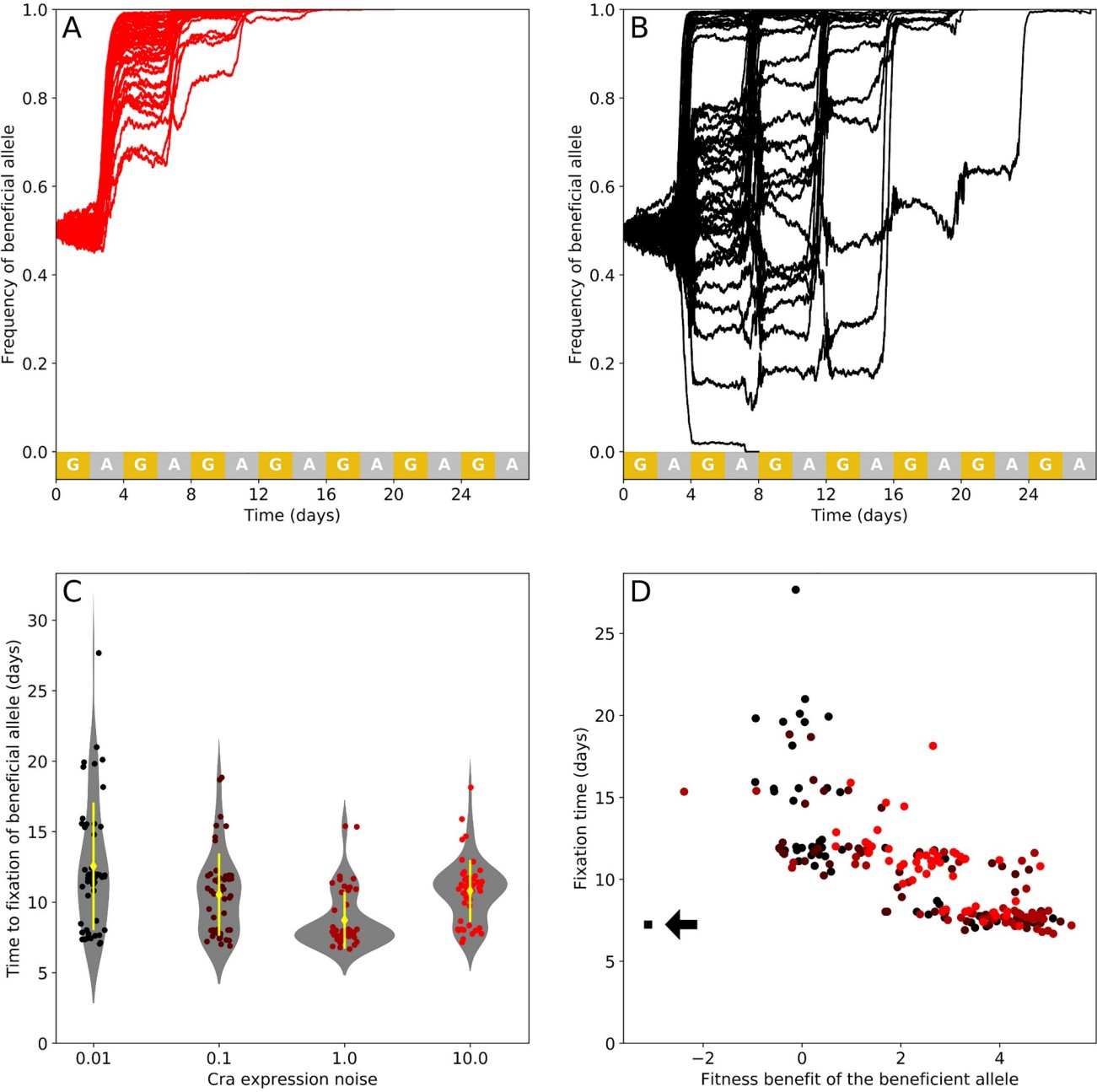

**Fig 5. A beneficial allele goes to fixation faster in populations with high Cra expression noise.** (A-B) Frequency of a beneficial allele competing with another allele in an environment that switches between glucose (yellow) and acetate (grey) every two days. The Cra dissociation constants have values of 0.07 and 0.05 mmol g$^{-1}$ for the beneficial and the wild-type Cra allele respectively. Both alleles occur in cells that have the same Cra expression noise, which is given by $\eta^2 = 10$ in (A), and $\eta^2 = 0.01$ in (B). Each plot show trajectories for 50 replicate population simulations. (C) Time until fixation of the beneficial allele (N = 50 replicates except for $\eta^2 = 0.01$, N = 49). Yellow diamonds and lines show the sample mean and one standard deviation. (D) The time the beneficial allele requires to go to fixation decreases with increasing fitness relative to its competitor (Spearman's $\rho = -0.77$, $p < 0.001$). Circles are coloured according to Cra expression noise. The square (arrow) marks the one replicate where the fitter allele was lost from the population.

## Beneficial alleles are more likely to invade when Cra expression noise is high

In the preceding section we asked whether gene expression noise can affect the competitive ability of a beneficial allele that occurs at a high frequency. However, most such alleles

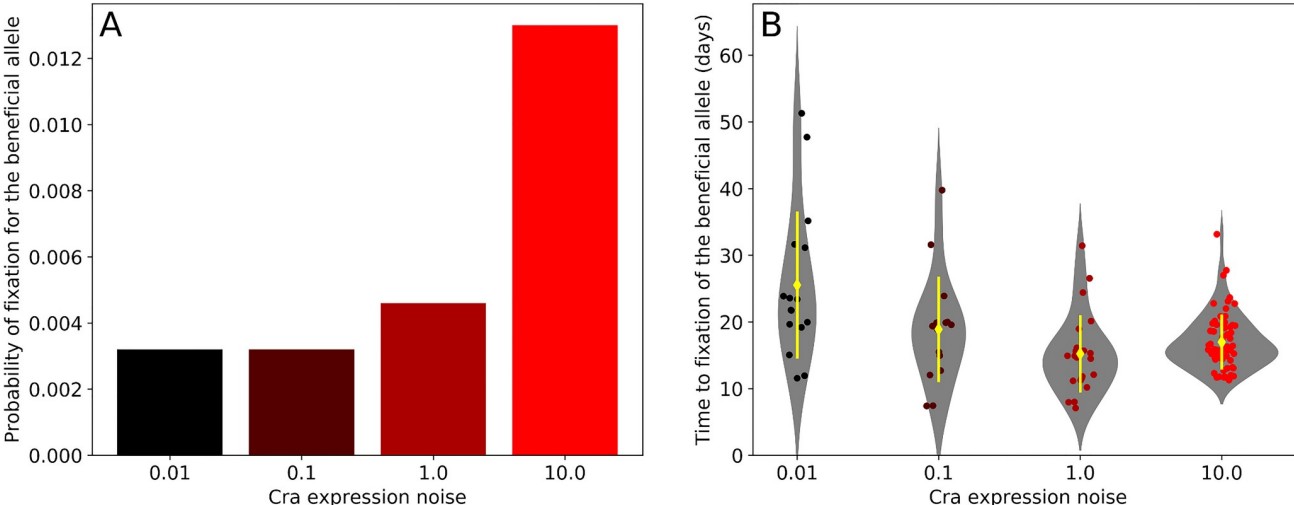

**Fig 6. Cra expression noise increases the probability of fixation of a beneficial allele that is initially at a low frequency.** We simulated populations that started with a single cell carrying a beneficial allele (dissociation constant of 0.07 mmol $g^{-1}$) in a population of 2000 cells. The other cells carried an allele that had a lower fitness (dissociation constant of 0.05 mmol $g^{-1}$). The population grew in an environment that switched between glucose and acetate every two days, and simulations continued until one of the two alleles was lost. We repeated these simulations 5000 times for each level of Cra expression noise. (A) The beneficial allele is more likely to go to fixation in populations with high Cra expression noise. (B) If the beneficial allele reached fixation, higher Cra expression noise decreases the time the beneficial allele needs to reach fixation, except for the highest level of noise (10.0). Yellow diamonds show the average time to fixation and yellow lines indicate one standard deviation. Circles show the fixation time of the beneficial allele in each replicate where it reached fixation. Number of observations from lowest to highest Cra expression noise: 16, 16, 23, 65.

originate in a single individual. We thus also wanted to find out whether such alleles can invade a population and go to fixation when initially rare. Therefore, we repeated our simulations, but starting from a population with 2000 individuals where only one individual carried the beneficial allele.

We found that when the initial frequency of the beneficial allele is low, it is most likely to go to fixation in populations with the highest Cra expression noise, with the observed probability of fixation increasing from 0.003 to 0.013 in the least noisy to the noisiest populations (Fig 6A, two-sided Fisher's exact test, $p < 0.001$, 5000 replicates). This increase in fixation probability cannot be caused by the allele's fitness on its own. If that were the case, the allele would be most likely to fix in populations with the second highest Cra expression noise, because that is where the fitness of a beneficial allele is highest (Fig 4C), and consequently the average time to fixation times is the lowest (Fig 6B).

Instead, the increase in fixation probability results from a decreased risk of elimination in populations with high Cra expression noise, because these populations experience less severe bottlenecks in acetate. For example, the average smallest population size during a population's first exposure to acetate is 37 cells (± 7 cells) and 475 cells (± 135 cells) for the populations with the lowest and highest Cra expression noise respectively. As a result of this difference, the number of simulations in which the beneficial allele survives the first environmental shift in populations with the lowest noise is less than half of that of populations with the highest noise (70 and 163 respectively). We observed an analogous difference for neutral alleles (see S1 Appendix section 10).

In sum, Cra expression noise favours the fixation of a novel and beneficial allele in two ways. First, greater Cra expression noise increases population size, thus decreasing the probability that a new allele is lost through drift. Second, it increases the fitness benefit derived from the new allele in a selective bottleneck, causing selection to help spread this allele faster. Both

of these effects can also be observed, albeit to a lesser extent, in environments that fluctuate randomly instead of periodically between glucose and acetate (see S1 Appendix section 11).

## Discussion

We present a stochastic, agent-based simulation of how a continually growing population of *E. coli* responds to a shift in carbon source from glucose to acetate. Each cell harbours a regulatory circuit that controls the metabolic switch between glycolysis, which cells use in the presence of glucose, and gluconeogenesis, which they use in the presence of acetate. The circuit is controlled by the transcription factor Cra [28, 29]. Cra activates enzymes necessary for growth on acetate [31, 32], and is inhibited during growth on glucose [24] by fructose-1,6-bisphosphate (fbp) [33]. When the sole carbon source changes from glucose to acetate, fbp levels drop and Cra activity may rise sufficiently for cells to start metabolising acetate. Kotte et al. [24] reported that, because the circuit behaves stochastically, only some cells grow on acetate while others do not. In other words, there is a bimodal distribution of growth rates on acetate. Consequently, a switch to acetate results in a lag phase at the population level, during which population growth appears to stop until the growing cells constitute a substantial fraction of the population. Upon return to glucose, all cells resume growth and the distribution of growth rates is unimodal. Cells that grew on acetate did not have a subsequent disadvantage on glucose [24]. Although this circuit embodies a specific regulatory mechanism, similar circuits that sense external conditions through intracellular metabolite concentrations are ubiquitous in bacteria [31]. Such ubiquity suggests that our observations may also be relevant for environments fluctuating in other carbon sources, such as cellobiose [25] or lactose [43].

Our aim was to find out how a source of nongenetic phenotypic variation, namely noise in the expression of the transcription factor Cra, affects the spreading of beneficial alleles in a fluctuating environment. We investigated a situation in which shorter lag times confer a fitness advantage, a scenario where previous studies have reported that gene expression noise can be beneficial [27, 44]. We caution that our results are valid only for cases where selection favours decreasing, not increasing lag time. In addition, our circuit captures only an aspect of lag times, namely heterogeneity in a regulatory circuit, and can only simulate lag times arising from a change in carbon source. Many bacteria regularly encounter feast and famine conditions [45, 46], and lag times commonly occur after long periods of starvation. The underlying regulation and physiology of these lag times are distinct from the case we have simulated here [45, 47]. See Bertrand [48] for a more general discussion of lag times and the selection pressures influencing them.

We showed that changing the expression noise of Cra without changing its mean expression level affects a population's recovery after the only available carbon and energy source in the environment changes from glucose to acetate. Higher Cra expression noise leads to a few cells with high Cra expression. In these cells, the high concentration of Cra increases the probability of expressing enzymes required for growth on acetate. These cells are therefore more likely to initiate growth on acetate, thus shortening lag times at the population level. Indeed, during the transition from glucose to acetate populations experienced a bottleneck, but this bottleneck was less severe when Cra was expressed noisily. We showed that the relative fitness of a population is mainly determined by the severity of this bottleneck. Consequently, populations with high Cra expression noise experience a shorter lag time and have a higher fitness.

To find out how Cra expression noise may influence the spread and eventual fixation of a beneficial allele, we modeled a beneficial allele that decreases the binding strength between Cra and its inhibitor fbp, compared to another, wild-type allele. This reduction in Cra-fbp binding strength is beneficial because it increased Cra activity and therefore shortens lag times.

The most important finding of our paper is that Cra expression noise influences the spread and eventual fixation of this beneficial allele in two ways. First, it leads to faster spreading of the beneficial allele. Second, it increases the likelihood that the allele does not become lost during a population bottleneck. We discuss these observations in greater detail in the following two paragraphs.

Cra expression noise accelerates the spread of the beneficial allele as long as it magnifies the fitness difference between the beneficial and wild-type alleles. We found that this amplification of fitness effects occurs for all levels of Cra expression noise we investigated, but that it is most pronounced at an intermediate level of Cra expression noise. At the highest Cra noise level, noise substantially decreases the lag time for all subpopulations, regardless of whether a subpopulation is carrying the beneficial or wild-type allele. In other words, the fitness benefit derived from noise alone can be so large that the fitness effect of a beneficial mutation becomes small by comparison.

Consistent with existing theory [49], we also found that the relative fitness between beneficial and wild-type alleles is a strong predictor of the amount of time a beneficial allele needs to spread to fixation from an intermediate frequency (p = 50%). However, we do caution that this effect is most pronounced in environments that fluctuate periodically. When environments fluctuate randomly, this effect is attenuated (S1 Appendix section 11).

We also found that higher Cra expression noise increases the probability that a rare beneficial allele survives population bottlenecks. This increase in survival occurs because higher Cra expression noise makes bottlenecks less severe, and the beneficial allele is thus less likely to be eliminated from the population by chance. This effect also increases the eventual fixation probability of the beneficial allele and is robust in environments that fluctuate randomly. Fluctuations in population size are known to reduce the fixation probability of beneficial mutations [2, 50, 51], especially because reductions in population size can lead to an increase in the power of drift and rapid elimination of genetic variation [52]. Empirical work regularly shows faster adaptation in populations that undergo less severe bottlenecks [53–55]. Although many theoretical studies consider the effect of population bottlenecks on the fate of mutations [50], most make the assumption that these fluctuations in population size are caused by factors outside the control of the organism [51, 52], although there are exceptions [23, 56, 57]. One such exception is a previous theoretical study, which has found that phenotypic plasticity can decrease the severity of population bottlenecks, thus increasing standing genetic variation [23]. This observation suggests that nongenetic variation has the potential to dampen fluctuations in population size more generally.

We caution that our study only demonstrates an acceleration of adaptation through gene expression noise in the short term. For two reasons, this advantage may disappear in the long term. First, Cra expression noise only results in a fitness benefit when the mean population fitness is low. In other words, the fitness benefit of Cra expression noise decreases as populations increase in fitness during adaptive evolution. This pattern holds for nongenetic variation in general, and is consistent with prior theoretical [16, 17] and empirical work [36]. It occurs because in populations remote from a fitness optimum, random variation in phenotype is more likely to push some individuals closer to the optimum than in a well-adapted population. In the latter kind of population, most random variation will push individuals away from the optimum [16]. These considerations suggest that Cra expression noise will only accelerate the spread of beneficial mutations at the beginning of an adaptive process when populations have very low fitness.

A second reason why noisy Cra expression may only provide short-term benefits is that high Cra expression noise leads to rapid elimination of genetic variation, which may slow down adaptation in the long term. Previous theoretical studies show that nongenetic variation

can strengthen genetic drift [10, 11], which we also observed in simulations of two competing alleles that are selectively neutral, i.e. that do not lead to a change in fitness (see S1 Appendix section 10). Strong genetic drift generally leads to fast elimination of genetic variation, some of which may be adaptive. In other words, populations with a higher level of nongenetic variation may also have a lower level of standing genetic variation. A well-known result of theoretical population genetics is that the response of a population to selection on a given trait is predicted by the amount of standing genetic variation in that trait [3]. Consequently, a population with high levels of nongenetic variation may be less responsive to selection, and thus adapt slowly. These considerations suggest that, even though beneficial mutations are more likely to spread and reach fixation once they arise in a population with much nongenetic variation, the overall rate of adaptation in such a population may be low compared to a population with less nongenetic variation.

What is important for the mechanism we identified here is that the source of nongenetic variation must itself be heritable, so that successive generations generate similar amounts of random phenotypic variation whenever the environment changes. Fortunately, there is no shortage of such heritable sources of nongenetic variation. The most obvious one in bacteria is gene expression noise, because it is highly heritable [40, 58]. However, many other sources may play this role, including variation in epigenetic markers in unicellular eukaryotes [59] or cancer cells [60], nongenetic variation in the surface proteins of pathogens [42] such as the highly variable surface glycoproteins of trypanosomes [61], or phenotypic plasticity in multicellular organisms [62].

From a broader perspective, our study highlights two especially important and well known properties of population bottlenecks in fluctuating environments. The first is that the capacity to mitigate population bottlenecks can be an important part of adapting to fluctuating environments. The second is that a particular kind of bottleneck called a *selective* bottleneck can increase the rate of adaptation. In a selective bottleneck, cells have heritable differences in the probability of surviving, in contrast to a *nonselective* bottleneck, where all cells have the same probability of surviving. Because competition for passing through a selective bottleneck can be particularly intense, selective bottlenecks have great potential to accelerate the spread of beneficial alleles [42], an effect also observed in natural populations [63]. Both selective bottlenecks and bottleneck mitigation are best-studied for the transmission of pathogens. Pathogens experience a strong bottleneck during transmission, because only a few infectious agents are transmitted successfully between hosts or between tissues within a host [42, 57, 64]. What is more, pathogens that are more likely to invade a new host or that can decrease the severity of the transmission bottleneck can have a large fitness advantage over others [42, 56, 57]. Most nonpathogenic organisms also experience selective bottlenecks, because natural environments are heterogeneous and unpredictable [46, 63, 65, 66]. While such fluctuations are often random or involve feast and famine scenarios, others are periodic. Examples include the intensity of predation of lynxes on hares [67] and seasonal changes in food availability [68] or temperature [69].

## Conclusion

We show that nongenetic variation in the form of gene expression noise can increase fitness by shortening a population's lag time before the population resumes growth in a new environment. We identify how noise can on the one hand accelerate the spread of a beneficial allele by enhancing its fitness benefit in a selective bottleneck that occurs periodically, and on the other hand protect the beneficial allele from extinction during the bottleneck. We postulate that our findings are not limited to bacteria but can be extended to other organisms experiencing

periodic changes in their environment that are associated with a reduction in population size, for example through seasonality.

## Materials and methods

### Model description

Our model simulates a population of *E. coli* cells continuously growing under a constant rate of influx of nutrients and a constant efflux of spent medium and cells, as would take place in a well-mixed environment such as a chemostat [70]. The inflowing medium contains either glucose or acetate as the only source of carbon and energy. Glucose and acetate molecules are then either taken up by cells or eventually flushed out of the chemostat (the differential equations modelling this behaviour are given in S1 Appendix section 5). We assume that nutrient and metabolite concentrations change instantaneously throughout the growth environment. Because we assume that cells undergo random fluctuations in protein content, we simulate each cell in the population individually. The simulation proceeds in intervals of $\Delta t$, which we set to one minute, during which (i) intra- and extracellular metabolite concentrations and cell masses are updated, (ii) protein are produced at random, (iii) cells are randomly flushed out of the growth environment, and (iv) cells that have doubled their mass divide. Details of the model implementation, parameter values, and model variables are given in S1 Appendix, sections 1-5.

We make the simplifying assumption that, in a constant environment, all variation in a cell's specific growth rate arises from variation in protein amounts only. We assume that fluctuations due to stochasticity in (i) metabolite diffusion, (ii) transcription factor binding to DNA, (iii) protein-metabolite binding, and (iv) the activity of a given enzyme happen on such short time scales that they can effectively be ignored on the longer scale of a cell's lifetime [71]. In our model, random fluctuations in protein amounts propagate to the specific growth rate through fluctuations in the rate of the reactions these proteins catalyse, as Kiviet et al. [72] observed experimentally in *E. coli*. Thus, still assuming a constant environment, these considerations entail that in our model all average reaction rates remain constant with fluctuations in protein amounts as the only source of variation. Our model implements this constancy in two ways. First, we simulate all processes except protein production [73], protein partitioning during cell division [74] and protein degradation [34] deterministically in a system of ordinary differential equations (ODE). Second, we increase the average amount of any protein per cell at the same rate as the cell mass to avoid a drop in reaction rates towards the end of the cell cycle due to protein dilution.

**Metabolism and growth: ODE system.** We implement metabolism as a system of ODEs that determines the temporal change of the extracellular acetate and glucose concentration, and of the intracellular metabolite concentrations of each cell. The model (Fig 1A) condenses glucose and acetate assimilation, glycolysis, its inverse gluconeogenesis, and cell growth into four metabolic reactions, each catalysed by a distinct enzyme [24]. Glucose uptake and conversion into fbp are condensed into glucose incorporation (Gi). Acetate uptake and conversion into pep through the glyoxylate shunt [75] is represented by acetate incorporation (Ai). Pep is then converted to fbp through lower glycolysis (Lg). Finally, fbp is converted to biomass through anabolism (An), whose kinetics are modelled after the enzyme that removes fbp from lower glycolysis in *E. coli*, fructose-bisphosphatase [32]. As a result, we need to model only four metabolite concentrations, that of glucose, acetate, pep and fbp.

In our model, both glucose and acetate are converted into fbp with the stoichiometry of the corresponding reactions in *E. coli* [32, 75], namely 1:1 for glucose to fbp, and 4:1 for acetate to fbp. One carbon atom is lost as $CO_2$ for every two acetate molecules that are converted to pep

(half circle in Fig 1A). The pathway from acetate to fbp goes through the intermediate pep with a stoichiometry of 2:1 for both acetate to pep, and pep to fbp. Fbp then gets converted into biomass with a yield of 0.0896 g of cell mass (dry weight) per mmol of fbp. We estimated the yield using Flux Balance Analysis (FBA) and the iJO1366 reconstruction of the *E. coli* metabolic network [76] with fbp as the only carbon source (with an influx of 1 mmol g$^{-1}$ h$^{-1}$). FBA is a constraint-based modelling tool that allows the prediction of reaction fluxes from a metabolic network and the flux of metabolites into a cell [77]. We use this yield as a coversion factor between the rate of fbp consumption and the specific growth rate and refer to it as *c*.

The biomass yield multiplied by the rate of the reaction converting fbp to biomass determines the specific growth rate $\mu$, i.e. the growth rate per unit cell mass. Cell mass increases exponentially, and the absolute growth rate of a cell is given by

$$\frac{\mathrm{d}B}{\mathrm{d}t} = \mu \cdot B \tag{1}$$

where $B$ is the current mass of the cell in grams dry weight, and the temporal derivative $\frac{\mathrm{d}B}{\mathrm{d}t}$ is given in g h$^{-1}$. We assume that newly divided cells have a mass of $B_0 = 3 \times 10^{-13}$ g, or 300 fg [78, 79], and divide once their mass has doubled. To keep the model as simple as possible, we ignore the scaling between growth rate and initial biomass [80].

For simplicity, we assume Michaelis Menten kinetics for all reactions except for the reaction that converts fbp into biomass, which follows Monod Wyman Changeux (MWC) kinetics [81] and is allosterically activated by pep, as experimentally observed [33]. The intracellular concentrations of pep and fbp are simulated per gram of biomass (mmol g$^{-1}$). The extracellular concentration of glucose or acetate is given in mM and is converted to mmol g$^{-1}$ when either carbon source is imported into the cell. The change in intracellular metabolite concentrations for each cell is determined by the rate $J_x$ of the four metabolic reactions $x$ (e.g. $J_{Gi}$ is the rate of the glucose incorporation reaction Gi) and the specific growth rate $\mu$, because metabolite concentrations undergo exponential decay as a cell grows unless they are replenished by an upstream reaction. Overall, these considerations lead to the following system of differential equations, which specify the change in intracellular metabolite concentrations for a given cell as

$$\frac{\mathrm{d}pep}{\mathrm{d}t} = \frac{1}{2}J_{Ai} - J_{Lg} - \mu \cdot pep \tag{2}$$

$$\frac{\mathrm{d}fbp}{\mathrm{d}t} = J_{Gi} + \frac{1}{2}J_{Lg} - J_{An} - \mu \cdot fbp, \tag{3}$$

where the coefficients on the right side reflect the stoichiometry of these reactions in *E. coli* [32, 75]. For example, to achieve a production rate of 1 mmol g$^{-1}$ h$^{-1}$ of fbp, the reaction rate of glucose incorporation $J_{Gi}$ would need to be 1 mmol g$^{-1}$ h$^{-1}$, while the reaction rate through lower glycolysis $J_{Lg}$ would have to be 2 mmol g$^{-1}$ h$^{-1}$. Details of the equations determining the reaction rates $J_{Ai}$, $J_{Gi}$, $J_{Lg}$, and $J_{An}$ are given in S1 Appendix section 5.

**Cra-fbp binding and expression of the acetate incorporation enzyme.** In *E. coli*, the transcription factor Cra regulates the switch from glycolysis to gluconeogenesis [24, 28, 31, 82, 83]. Cra is inhibited by fbp [84]. As long as glucose is fed to a cell, fbp levels remain high and Cra activity is suppressed. Once a cell is starved of glucose, the fbp concentration falls, and the increased level of Cra activity promotes the expression of genes involved in acetate assimilation and gluconeogenesis [24, 28, 29].

We model this process by having Cra activate the transcription of the acetate consumption pathway, which we represent by the enzyme Ai. Cra targets the expression of enzymes at the

entry and exit points of lower glycolysis, which are irreversible reactions producing and consuming fbp and pep, but not of the reversible reactions in lower glycolysis between fbp and pep [82, 83]. We therefore kept the expression of the enzyme converting pep to fbp independent of Cra activity. Cra repression by fbp follows Hill kinetics [24], which we model in the following way

$$Cra_A = Cra \times \frac{K_{Cra,fbp}{}^{n_{Cra}}}{fbp^{n_{Cra}} + K_{Cra,fbp}{}^{n_{Cra}}} \tag{4}$$

where $Cra_A$ is the number of active Cra protein, $K_{Cra,fbp}$ is the dissociation constant for Cra-fbp binding, and $n_{Cra}$ is the Hill coefficient governing the cooperativity of binding. We model $Cra_A$ as a continuous rather than a discrete variable because we average Cra activity over the interval $\Delta t$. Specifically, although the number of active Cra proteins at any given time must be discrete, we assume that the random fluctuations between the active and inhibited forms occur rapidly enough to model Cra activity as a single continuous variable. For example, if a cell harbours a single Cra molecule, which is active for half the time interval $\Delta t$, then $Cra_A$ is 0.5.

We assume that the active form of Cra binds the Ai promoter with Michaelis Menten kinetics, as this is the simplest model for an activating transcription factor [71]. To account for dilution during cell growth, we use the $Cra_A$ concentration ($Cra_A$ activity divided by current cell mass $B$) to determine the proportion of time $P_{bound}$ that $Cra_A$ is bound to the promoter. It is given by

$$P_{bound} = \frac{Cra_A/B}{Cra_A/B + K_{Cra_A,DNA}} \tag{5}$$

where $K_{Cra_A,DNA}$ is the dissociation constant of $Cra_A$ for the Ai promoter. $P_{bound}$ then determines the expected mRNA production rate per protein half-life $\alpha_{Ai}$, which is the time average of two rates, namely the mRNA production rate when $Cra_A$ is bound ($\alpha_{Ai,1}$), and unbound ($\alpha_{Ai,0}$, which we set to zero to represent an "off" state).

$$\alpha_{Ai} = \alpha_{Ai,0} \cdot (1 - P_{bound}) + \alpha_{Ai,1} \cdot P_{bound} \tag{6}$$

We use $\alpha_{Ai}$ to model the stochastic production of Ai in the same way as for all other proteins.

**Protein production, degradation and dilution.**   Because noise in protein concentrations arises in part from low protein numbers, we simulate the number of proteins per cell explicitly. We use a simple model of protein production and degradation, which assumes that proteins are expressed from a constitutively expressed gene [85]

$$DNA \xrightarrow{k_1} mRNA \xrightarrow{k_2} Protein$$
$$\quad\quad\quad \downarrow \gamma_1 \quad\quad\quad \downarrow \gamma_2$$
$$\quad\quad\quad \varnothing \quad\quad\quad\quad \varnothing$$

where $k_1$ and $k_2$ are the transcription and translation rates, and $\gamma_1$, $\gamma_2$ the mRNA and protein degradation rates. This model assumes that transcription, translation, and mRNA degradation are Poisson processes. Ribosomes and ribonucleases compete for mRNA, resulting in a geometric distribution of protein molecules synthesised from a single mRNA [86]. In consequence, the steady-state distribution of protein numbers $X$ per cell follows a negative binomial distribution [85, 86]

$$X \sim NegBin(\alpha, \beta) \tag{7}$$

where $\alpha = k_1/\gamma_2$ can be interpreted as the expected number of mRNA transcripts produced

during a protein's half-life, and $\beta = k_2/\gamma_1$ is the expected number of proteins produced from a given mRNA. The mean of the distribution is given by $\bar{x} = \alpha\beta$, and the variance by $\sigma^2 = \alpha\beta(\beta + 1)$. Protein expression noise ($\eta^2 = \sigma^2/\bar{x}^2$) is equal to the inverse of the mRNA production rate $1/\alpha$. These quantities agree well with single-cell observations [37, 87]. For convenience, we refer to the average number of proteins per cell ($\alpha\beta$) as the number of proteins in a newborn cell, so that a cell that is about to divide has twice as many proteins on average ($2\alpha\beta$). By tuning the parameters $\alpha$ and $\beta$, we can change the variance of protein expression without modifying the mean. In other words, we can change expression noise independently from mean expression.

Because our modelling approach aims to keep the average number of proteins per unit cell mass constant, but our model allows the growth rate of cells to vary, we couple the protein production rate to the growth rate. Variation in the growth rate affects the rate of dilution of proteins, which we subsume under the protein degradation term $\gamma_2$, as in [37, 85]. We take the protein degradation term $\gamma_2$ to be the sum of the active protein degradation rate $\gamma_p$ and the protein dilution rate $\gamma_d$, i.e. $\gamma_2 = \gamma_p + \gamma_d$. The protein dilution rate $\gamma_d$ is set by the absolute growth rate $\mu \cdot B$ (Eq 1), because cell growth is exponential and consequently proteins are diluted twice as fast at the end of the cell cycle than at the beginning (given a constant specific growth rate $\mu$). Because the protein dilution rate $\gamma_d$ is given in units of time ($h^{-1}$), we divide the absolute growth rate by the initial cell mass $B_0$, so that $\gamma_d = \mu \cdot B/B_0$. Therefore, the protein dilution rate is equal to $\mu$ at the beginning of a cell cycle, and $2\mu$ at the end of a cell cycle. To keep the average number of proteins per unit cell mass constant, the protein production rate must stay in step with the rate of protein dilution and degradation. The protein degradation term $\gamma_2$ affects the average protein number ($\alpha\beta$) through $\alpha$, thus if $\alpha$ is to remain constant, the transcription rate $k_1$ has to compensate for fluctuations in $\gamma_2$. Taking these considerations together leads us to the following expression for the rate of transcription $k_1$ for each protein at each point in time:

$$k_1 = \alpha \cdot \left( \mu \cdot \frac{B}{B_0} + \gamma_p \right) \qquad (8)$$

Because most proteins in *E. coli* undergo little to no active degradation [34], we performed most of our simulations with $\gamma_p = 0$. The half-lives of stable proteins in *E. coli* range between about 1 hour under starvation conditions to over 70 hours (BNID 109921, [88]). In other words, degradation rates range from about $\gamma_p = 0.7$ to $0.01$ $h^{-1}$ or lower. Increasing the degradation rate increases the sensitivity of a protein's amount to changes in its production rate [71]. Because the acetate incorporation enzyme Ai is the only protein whose production rate changes in response to the environment, we only investigated active degradation of Ai, which we modelled as a process of stochastic decay. Specifically, the number of proteins remaining after a given time interval $\Delta t$ follows a binomial distribution in which the probability of survival decays exponentially at the rate $\gamma_p$.

## Estimating fitness

To compare the fitness of different populations, we conducted simulations in which we competed each population of interest against a reference population with an intermediate level of noise $\eta^2 = 0.2$ and either the same or an intermediate Cra-fbp binding strength (when we investigated multiple Cra-fbp dissociation constants, as in Fig 3B). All simulations started with both populations at an initial size of 1000 cells. The competing populations were first exposed for two days to glucose, and then for an additional two days to acetate. We estimated the relative fitness of the two competing populations by calculating the change in frequency of the

population of interest during the simulation [89]

$$w = \ln \left( \frac{N_1/N_0}{N_1'/N_0'} \right) \tag{9}$$

where $N_0$ and $N_1$ are the numbers of cells of the population of interest at the beginning and the end of the simulation, respectively. Similarly, $N_0'$ and $N_1'$ are the number of cells of the reference population at the beginning and the end. If one defines the fitness of the reference population as equal to one, this measure becomes equivalent to the selection coefficient of the population of interest compared to the reference population [90]. The measure creates a fitness scale in which a population that is fitter than the reference population will have a positive relative fitness, while a population with lower fitness will have a negative relative fitness.

## Supporting information

**S1 Appendix. Additional model details and further analysis.** This appendix shows parameter values, variables, and further details about the model and its implementation. It presents an analysis of how a growing and a nongrowing state can arise in a deterministic version of the model, and estimates how robust this bistability is against changes in parameter values. It also considers how Cra amounts influence the growth state a cell is in. In addition, it contains simulations showing the effect of active degradation of the acetate incorporation enzyme. Finally, it considers the spread of a neutral allele in populations with different levels of Cra expression noise and concludes with simulations of randomly fluctuating environments.
(PDF)

## Acknowledgments

MS would like to thank Jordi van Gestel and Eugenio Azpeitia for thoughtful conversations on bacterial evolution and gene expression, and Joseph Schmutzer and Tess Brewer for their comments on the manuscript.

## Author Contributions

**Conceptualization:** Michael Schmutzer, Andreas Wagner.

**Formal analysis:** Michael Schmutzer, Andreas Wagner.

**Funding acquisition:** Andreas Wagner.

**Methodology:** Michael Schmutzer.

**Project administration:** Andreas Wagner.

**Supervision:** Andreas Wagner.

**Writing – original draft:** Michael Schmutzer, Andreas Wagner.

**Writing – review & editing:** Michael Schmutzer, Andreas Wagner.

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
