## [Decision Letter · Decision Letter 0]

27 May 2020

Dear Mr Schmutzer,

Thank you very much for submitting your manuscript "Gene expression noise can promote the fixation of beneficial mutations in fluctuating environments" for consideration at PLOS Computational Biology.

As with all papers reviewed by the journal, your manuscript was reviewed by members of the editorial board and by several independent reviewers. In light of the reviews (below this email), we would like to invite the resubmission of a significantly-revised version that takes into account the reviewers' comments. 

We cannot make any decision about publication until we have seen the revised manuscript and your response to the reviewers' comments. Your revised manuscript is also likely to be sent to reviewers for further evaluation.

Sincerely,

Mark Goulian

Guest Editor

PLOS Computational Biology

Mark Alber

Deputy Editor

PLOS Computational Biology

Reviewer's Responses to Questions

**Comments to the Authors:**

Reviewer #1: This paper uses a mechanistic model of stochastic switching between glucose and acetate metabolism in a microbe to explore the effects of gene expression noise on both fitness and evolvability. I think this topic is ripe for modeling contributions of this sort and also perfect for the journal. The modeling approach and the evolutionary analysis both seem good, though there is some room for polishing of the presentation.

I was quite confused about the construction of beneficial alleles--these were introduced as "confer[ring] a fitness advantage that is independent of Cra expression noise", but then Fig. 4 demonstrates how this benefit is not independent of expression noise. (Also, Fig. 4 could benefit from more replication; it's hard to get much out of panel B). It was hard to follow the set-up for this section--the hypothesis that noise must boost adaptation rate by boosting fitness effects seemed to come out of nowhere. The results that follow are interesting but somewhat hard to link with typical evolutionary theory results in simpler models. The benefits of noise for selection of a beneficial mutation seem to be a complicated blend of increasing geometric mean fitness (or just increasing fitness in the acetate environment without cost in glucose) and within-generation bet-hedging following a bottleneck. I would like to see this half of the paper clarified quite a bit, with a more clear set-up and better connection with the evolutionary literature on bet-hedging. This criticism is mitigated by the clear differentiation of these different effects in the Discussion; it is mostly in the Results that I would suggest revision.

Overall, I think this is a strong contribution that will interest readers in microbiology, systems biology, and evolution, and would be suitable for publication after a minor revision.

Notes (written as I read to highlight points of confusion)

Lines 15-26: There's a broader literature here that you might want to reference based on nongenetic phenotypic heterogeneity smoothing out fitness landscapes. I'd recommend Frank (2011) JEB and Whitlock (1997) Evolution.

Lines 72-74: Is this really a prediction, or a premise of the way you designed the model?

Fig. 1B: Could you add a small plot of mean fop and pep concentrations for each bin of growth rate on acetate? It's not strictly necessary but might help reinforce the text.

Fig. 2A: This didn't render well for me; can't distinguish the four treatments. Recommend rethinking colors and approach here.

Line 202: Could there be costs of noise during growth on glucose in your model? If so, I would prefer to measure both the costs and benefits by teasing them apart; if not, then it seems clear there will be a benefit of higher noise and I would change the focus to quantifying them, rather than reporting ANOVA results.

Lines 238-239: This is too narrow, as there are several distinct ways that expression noise could affect the rate of adaptation. Additionally, noise could complement mutations that affect any life-history trait, not just lag reduction--were more generic types of beneficial mutations considered? So, to me this is too strict in two dimensions. The section title also hints that you're going in a different direction--changing effective population size rather than the fitness benefit. Or, is this going to be about geometric mean fitness--but that's not what "ratio of bottleneck sizes" invokes for me. Overall, I found that the paper gets quite confusing in this paragraph.

Reviewer #2: Using agent based simulations, Schmutzer & Wagner investigate the effect of noise in Cra expression on the evolutionary dynamics and fitness of a population of E.coli that undergoes a change from exponential growth on glucose to growth on acetate. They show that the presence of noise in the repressed state of Cra (under glucose conditions) is beneficial for bacteria when they switch to acetate, as this leads to higher fitness of the population and a decrease in bottlenecks due to the fact that Cra is repressed in the initial condition.

My main points of criticism are:

1. The study considers only switching from one condition to the other (glucose to acetate) after being in very long term exponential growth, i.e. chemostat conditions. Such conditions are really the exception in nature, as most microbes live under repeated feast and famine conditions, namely cells in stationary phase will encounter a new type of nutrient. Hence one should be rather careful in generalizing the results obtained from such idealized initial setup.

2. The authors consider switching events between the two food sources in fluctuating environments that are periodic, again not very typical in nature. In order to have a more general view of the effect of noise on fitness in this specific system, especially given the fact that Cra it is a global regulator, it would be helpful to explore how the populations fare in non-periodic fluctuating environments, namely with variable switching times between environments. Given that the results are simulations based and do not rely on analytical solutions this I find crucial.

3. The numbers of Cra molecules used for the stochastic part of the model (lines 149 to 155 or so) do not seem to come from experimental determinations of Cra molecules in the cell. A mean of 100 molecules seems low for a global regulator.

Thus, the generality claims need to be brought down a notch. I think we need to be a bit more careful with what noise does or does not do for biology and evolution, as we risk to fall into the trap of the Spandrels of San Marco again, by ignoring the wider context of how organisms evolve, in this case the feast and famine aspect and truly fluctuating nature of this dynamics as opposed to cyclic scenarios.

A few more minor things:

The text in general can be cut down as many things can be explained in a more brief manner. It is quite tedious to read in places.

The use of yellow in several of the figures makes it hard to read the figures.

Reviewer #3: Attached.

**Have all data underlying the figures and results presented in the manuscript been provided?**

Reviewer #1: Yes

Reviewer #2: Yes

Reviewer #3: Yes

PLOS authors have the option to publish the peer review history of their article (what does this mean?). If published, this will include your full peer review and any attached files.

Reviewer #1: No

Reviewer #2: No

Reviewer #3: No
---

## [Decision Letter · Decision Letter 1]

15 Sep 2020

Dear Mr Schmutzer,

We are pleased to inform you that your manuscript 'Gene expression noise can promote the fixation of beneficial mutations in fluctuating environments' has been provisionally accepted for publication in PLOS Computational Biology.

Best regards,

Mark Goulian

Guest Editor

PLOS Computational Biology

Mark Alber

Deputy Editor

PLOS Computational Biology

Reviewer's Responses to Questions

**Comments to the Authors:**

Reviewer #1: The authors have addressed all my comments very thoroughly. I have read the other reviews and the revised paper and see no further cause for revision. I think this will make a valuable contribution.

Reviewer #2: The authors addressed my main and minor concerns. Am glad to see that they did do the simulations in fluctuating environments and that these are indeed different from those in regularly switching ones. It is always worth checking....

Am also happy to see that the generality if the message was toned down.

Now it is a more honest and solid manuscript.

**Have all data underlying the figures and results presented in the manuscript been provided?**

Reviewer #1: Yes

Reviewer #2: None

PLOS authors have the option to publish the peer review history of their article (what does this mean?). If published, this will include your full peer review and any attached files.

Reviewer #1: No

Reviewer #2: No

---

## [Editor Report · Acceptance letter]

16 Oct 2020

PCOMPBIOL-D-20-00200R1 

Gene expression noise can promote the fixation of beneficial mutations in fluctuating environments

Dear Dr Schmutzer,

I am pleased to inform you that your manuscript has been formally accepted for publication in PLOS Computational Biology. Your manuscript is now with our production department and you will be notified of the publication date in due course.

With kind regards,

Sarah Hammond
